# Prehabilitation in Cardiovascular Surgery: The Effect of Neuromuscular Electrical Stimulation (Randomized Clinical Trial)

**DOI:** 10.3390/ijerph20032678

**Published:** 2023-02-02

**Authors:** Alexey N. Sumin, Pavel A. Oleinik, Andrey V. Bezdenezhnykh, Natalia A. Bezdenezhnykh

**Affiliations:** Laboratory for Comorbidity in Cardiovascular Diseases, Federal State Budgetary Institution “Research Institute for Complex Issues of Cardiovascular Diseases”, 6, Sosnoviy blvd, 650002 Kemerovo, Russia

**Keywords:** neuromuscular electrical stimulation, prehabilitation, cardiovascular surgery, muscle status

## Abstract

Objective: We aimed to determine the effects of prehabilitation with neuromuscular electrical stimulation (NMES) on muscle status and exercise capacity in patients before cardiac surgery. Methods: Preoperative elective cardiac surgery patients were randomly assigned to the NMES group or control group. Intervention in the NMES group was 7–10 sessions, whereas the control group carried out breathing exercises and an educational program. The outcome measures included a six-minute walk test (6MWT) and a muscle status assessment (knee extensor strength (KES), knee flexor strength (KFS), and handgrip strength (HS)) after the course of prehabilitation. Results: A total of 122 patients (NMES, n = 62; control, n = 60) completed the study. During the NMES course, no complications occurred. After the course prehabilitation KES, KFS, and 6MWT distance were significantly increased (all *p* < 0.001) in the NMES group compared to the control. There was no significant difference in HS before surgery. Conclusions: A short-term NMES course before cardiac surgery is feasible, safe, and effective to improve preoperative functional capacity (six-minute walk distance) and the strength of stimulated muscles.

## 1. Introduction

Due to an ageing population, the number of elderly patients undergoing heart surgery is increasing. The risk of such an operation is higher in elderly patients, therefore all risk assessment scales include age, but such assessments are incomplete and are not able to accurately predict the risk of surgery in clinical conditions. Since biological age cannot characterize the degree of the patient’s loss of function, it is proposed that the concept of “frailty” is used to reflect this parameter [1]. The problem of weakness before cardiac surgery is not uncommon; the frequency of its detection usually varies from 20% to 50% in different studies [2,3]. Using a broader concept, pre-fragility was present in 65.2% of patients undergoing elective coronary artery bypass grafting [4]. When using the assessment of “frailty” and “prefrailty” in the clinic, it was found that these indicators are independent prognostic factors for heart surgery [5,6]. For example, patients with prefrailty have longer ventilatory times, longer ICU and hospital stays, and complications, such as stroke or in-hospital death, when compared to non-frailty patients after cardiovascular surgery [4]. Often, “frailty” is not fully evaluated by clinicians; they determine only the patient’s muscular status. There are approaches with an anatomical (assessing the cross-sectional area of muscles using MRI or ultrasound [7]) or a functional (evaluating muscle strength and endurance) assessment of muscle status.

Studies have shown that these methods also allow you to further assess the risk of cardiac surgery [7,8,9]. In addition, fragility is not only a reflection of age-related changes but also occurs at a younger age (for example, up to 11.6% at the age of 50–64 years [10]). The decrease in muscle status can occur not only due to age-related changes but also due to a decrease in physical activity due to an underlying cardiological disease (“secondary sarcopenia”).

As a result, researchers propose that, in the decision-making process on the possibility of cardiac surgery in elderly patients, the frailty and muscle status, and not the age of the patient, are taken into account [11,12]. Another conclusion from the presented studies is that it is advisable to use methods to improve the physical status of patients before cardiac surgery, but the results of such studies are contradictory. Local respiratory muscle training and breathing exercises prior to cardiac surgery reduced the risk of developing postoperative pneumonia, postoperative atelectasis, and reduced the postoperative hospital stay [13]. In a recent meta-analysis of six studies, it was shown that exercise (respiratory muscle training, aerobics, strength training, and stretching) can help patients recover from cardiac surgery. In the exercise group, the length of stay in the intensive care unit and the postoperative physical function were improved compared to the control group. However, no significant differences were observed in the incidence of postoperative complications and cognitive function [14]. Nevertheless, the utilization of conventional rehabilitation programs in patients before cardiac surgery may be difficult due to the severity of the underlying cardiac disease that limits the patient’s physical activity. In this regard, conducting local physical training using neuromuscular electrical stimulation (NMES) may be appropriate [15,16,17,18]. Ivatsu et al. showed that the use of NMES is safe in patients in the early stages after cardiovascular surgery and that NMES also reduces proteolysis and skeletal muscle weakness [15,16]. However, at the preoperative stage, NMES has not yet been used. Therefore, a pilot randomized controlled trial was performed to examine the effects of prehabilitation with NMES on muscle status in patients before cardiac surgery.

## 2. Materials and Methods

### 2.1. Study Design and Participants

This clinical randomized study was performed in the cardiology department of the Research Institute for Complex Issues of Cardiovascular Diseases, Kemerovo. Consecutive patients who underwent in-patient examination and preparation for cardiac surgery from 7 September 2020 to 30 September 2022 were approached. Exclusion criteria for the study included: surgical interventions in an emergency and urgent manner; arthropathies that prevented the complete and painless performance of the physical fitness test; low pain threshold; rhabdomyolysis and other myopathies; decrease or loss in cognitive function, which prevents full familiarization with the study protocol, age at the time of the survey being less than 35 and more than 80 years; and patient refusal to participate in the study. The study protocol was approved by the Local Ethics Committee of the Institution (Protocol No. 20170128) and was developed according to the Declaration of Helsinki, 2000 edition. Written informed consent was obtained from all patients prior to entering the study. This work was supported by exploratory scientific research “Preoperative preparation of patients with low exercise tolerance for cardiac surgery” (No 2020-419-32). The study protocol is registered on Clinicaltrias.gov (NCT04545268).

Eligible patients were randomly assigned to the NMES group or control group using a computer-generated randomization program. Patients in the NMES group underwent muscle stimulation daily before their cardiac surgery and the control patients received the usual pre-surgery program (breathing exercises and education).

### 2.2. Methodology

Upon admission to the hospital, patients underwent a standard preoperative examination, which included demographic, clinical, and biochemical parameters. Among the anamnestic indicators, the presence of myocardial revascularization, myocardial infarction, stroke, arterial hypertension, and other comorbid conditions (diabetes mellitus, chronic lung diseases, peripheral atherosclerosis) was taken into account. Patients also underwent ultrasound examinations (transthoracic echocardiography and duplex ultrasound examination of blood vessels).

Echocardiography was performed on “Vivid-7 Dimension” apparatus (General Electric, Boston, MA, USA) as per the current guidelines [19]. The thickness of the left ventricular (LV) wall was measured in a two-dimensional M-mode. The end-systolic and end-diastolic volumes of the LV and the maximum transverse diameter of the left atrium (LA) were evaluated. Left ventricular ejection fraction (LVEF) was calculated using the Simpson method. The diastolic function of the left ventricle was assessed by the E/A ratio; we also determined the systolic pressure in the pulmonary artery and the presence of stenosis and regurgitation of the heart valves.

Duplex ultrasonography was carried out with a 7.5 MHz linear-array transducer on “Vivid 7 Dimension” (General Electric, Boston, MA, USA) apparatus that followed standard imaging protocols. Doppler and B-mode, according to the degree of stenosis, measured the assessment of the narrowing of the carotid and lower extremity arteries.

We assessed the functional state and muscle status of patients before and after the course of prehabilitation. We examined muscle status by assessing the knee extensors and flexors and handgrip strength muscle groups. The maximum strength of the knee extensors and flexors was assessed using an isokinetic dynamometer “Lafayette MMT 01165” (Lafayette Instrument Company, Lafayette, IN, USA). The assessment of isometric muscle strength of the knee extensors and flexors was evaluated in a sitting position with the patients producing maximum muscle efforts, while the results were evaluated (maximum muscle strength) directly on the device screen. Exercises were performed in pairs for knee extensors and flexors. Hand grip strength (HS) was measured with a DK-100 dynamometer (RF) sequentially on the right and left upper limbs. Before the course of prehabilitation and after its completion, a 6 min walk test (6MWT) was performed in the corridor with a marked walking distance.

### 2.3. Intervention

The NMES course began on the second day of hospital stay after the muscle status and exercise capacity assessment. The duration of the NMES course was at least seven sessions (usually 7–10) daily, including the day before surgery and during the entire period of the patient’s stay in the hospital at the preoperative stage. The NMES methodology has been described in detail previously [18]. For NMES, Beurer EM80 apparatus (Beurer, Ulm, Germany) was used. The electrodes were placed over the attachment points of the quadriceps femoris muscle on the left and right limbs. The duration of each session was 90 min. With the help of rectangular pulses with a frequency of 45 Hz, a tonic contraction of M. quadriceps femoris was induced. The duration of each contraction was 12 s, with a pause between contractions of 5 s. The intensity of the electrical impulse was dependent on the patient’s pain threshold and was selected separately for each of the channels in the stimulator. In this case, it was necessary to achieve maximum muscle contraction, but without the occurrence of pain. In the control group, along with the preparation for the surgery, consisting of breathing exercises and an educational program, patients were recommended to adhere to their usual physical activity.

### 2.4. Statistical Analysis

Statistical analysis was performed using STATISTICA 10.0 software. The Shapiro–Wilk test was used to check the quantitative data for the type of distribution. Since the distribution differed from normal, quantitative variables are presented as median and lower and upper quartiles. The Mann–Whitney test was used to compare differences between the NMES group and controls. Categorical data are presented as the number of patients and percentage of the total sample and group comparisons were made using the χ^2^ test and Fisher’s exact test. The Wilcoxon test was used to assess changes in muscle status and 6MWT in the groups during prehabilitation. A two-tailed *p* value < 0.05 was used as the cut-off level for a statistically significant association.

## 3. Results

A flow diagram of the study participants is presented in Figure 1.

### CONSORT 2010 Flow Diagram

During the study period, 258 patients underwent inpatient examination and preparation for heart surgery in the clinic. Of these patients, 114 patients did not meet the inclusion criteria, 8 refused to participate in the study, and in 2 patients, after the examination, it was decided that surgery was not appropriate. As a result, 134 patients were included in this study and were randomized into two groups: the NMES group (n = 67), in which, along with the standard preoperative rehabilitation program, patients received a course of NMES, and the control group (n = 67), who underwent a standard preoperative rehabilitation program. For various reasons, 13 patients did not complete the prehabilitation course (5 in the NMES group and 7 in the control group), so 122 patients were included in the final analysis: 62 patients in the NMES group and 60 patients in the control group. When comparing the baseline parameters in the groups, there were no differences in clinical and anamnestic data (Table 1) and in the results of laboratory and instrumental studies (Table 2). 

The initial strength indicators of the studied muscles and 6MWT distance also had no significant differences in the groups (Table 3). 

The change in the maximal knee extensor strength (KES) from baseline to the second test after prehabilitation is shown in Figure 2. KES increased significantly in the NMES group, in contrast to the control group. Accordingly, KES in the second test was significantly higher in the NMES group than in the control (*p* < 0.001 on the right and the left legs).

The maximal knee flexor strength (KFS) increased from baseline to the second test only in the NMES group, not in the control (Table 3). Accordingly, KES in the second test was significantly higher in the NMES group than in the non-NMES group (*p* = 0.006 on the right and *p* = 0.005 on the left) (Figure 3).

The handgrip strength during the re-examination slightly increased in the NMES group, and in the control, it decreased slightly, but the differences in the re-test did not reach statistical significance (*p* = 0.054 on the right and *p* = 0.062 on the left; Figure 4, Table 3).

A similar dynamic was noted for the 6MWT—the distance during the repeated test in the NMES group increased, and in the control, it slightly decreased. This turned out to be sufficient to obtain statistically significant differences between the groups in the repeated test (*p* = 0.006) (Figure 5, Table 3).

Other indicators of muscle status are presented in Table 3. However, if in the second test, the average KES was significantly higher in the NMES group than in the group without NMES (*p* = 0.003 on the right and *p* = 0.002 on the left), then the average KFS in the groups did not differ (*p* = 0.116 on the right and *p* = 0.056 on the left).

## 4. Discussion

Our study showed that in the NMES group, compared with the control group, there was a more noticeable increase in the strength of the extensor and flexor forces of the knee joint and the distance of 6MT after a course of prehabilitation in preparation for cardiac surgery. However, we did not reveal the effect of NEMS on unstimulated muscles (handgrip strength). 

When used alone, NMES is inferior in its effect on muscle strength to conventional training with voluntary muscle contractions [20]; therefore, a combination of these two types of training is often considered [21], as is recommended for athletes [22]. When considering the physiological changes in muscles during NMES, it was found that during NMES, activation of muscle fibers that do not participate in voluntary muscle contraction occurs [23]. This suggests that adaptive processes in skeletal muscles affect a larger number of muscle fibers [24], which increases the performance of the muscle as a whole. Additionally, it is known that the phenomenon of neural adaptations (i.e., exercise-induced changes in nervous system function) results from the NMES program, which explains the effect of improving the muscle strength of contralateral muscle groups not involved in stimulation [25]. Previously, it was shown that intense modes of exposure to NMES have the maximum effect on stimulated muscles [26]; however, an alternative can be a sufficiently long course of stimulation (for example, for athletes—for 4 weeks). It can be assumed that a short 5-day course of NMES with a moderate degree of exposure in patients after cardiac surgery had a limited effect on muscle status due to insufficient intensity of exposure [17,27]. In the present study, we carried out a longer course of NMES with no effect of postoperative maladjustment for patients and no possible problems with postoperative wounds on the extremities. This may well explain the additional increase in stimulated muscle strength after NMES.

To the best of our knowledge, prior to this work, no attempt had been made to assess the effect of NMES on muscle strength prior to cardiac surgery. Research has primarily focused on the ability of NMES to improve muscle strength in patients after cardiovascular surgery [15,16,17,18,27]. However, these studies did not provide an unequivocal answer to the question of whether a course of NMES can improve muscle status in patients after cardiac surgery. For example, in a study by Iwatsu et al., the principal possibility of using NMES after such operations was shown [15]. In a subsequent study, these authors showed that NMES during the first five days after surgery led to an increase in muscle strength in certain muscle groups (an increase in knee extensor isometric strength and handgrip strength) [16]. However, in a more carefully planned randomized trial, an NMES effect on the strength of stimulated muscles was not found [17]. A feature of this study was that patients were included before surgery. This method of inclusion naturally led to most patients having an uncomplicated postoperative period, and patients participated in the rapid activation program. Consequently, the addition of a short course of NMES did not appear to be sufficient to further increase muscle strength, as shown by another study with a similar design [27]. On the contrary, with a longer stay in the ICU, as seen in the Catastim 2 study (mean time—6–7 days), muscle strength was restored faster in the NMES group [28]. According to the experience of our group, in patients with complications after cardiac surgery and a long stay in the ICU, a course of NMES also led to an increase in stimulated muscle strength [18].

To date, pre-rehabilitation programs before cardiac surgery have consisted of traditional outpatient physical training. With such programs, the goal was not to affect the muscular status of patients; instead, the main task was to reduce the number of perioperative complications and the results of operations in general. For example, a meta-analysis of eight small studies showed that preoperative physical therapy (including local training of the respiratory muscles) can reduce postoperative pulmonary complications (atelectasis and pneumonia) and the length of hospital stay in patients having elective heart surgery. At the same time, such pre-rehabilitation did not affect the incidence of postoperative pneumothorax, the duration of mechanical ventilation, or mortality from all causes [13]. These data show that even local training of individual muscle groups can favourably influence recovery after cardiac surgery. Moreover, as a recent study by Chen et al. [29] showed, even a five-day course of intensive preoperative training of inspiratory muscles reduced the incidence of postoperative pulmonary complications and the duration of postoperative hospitalization in patients undergoing heart surgery. Other possible types of training (aerobic training, static exercises) were practically not used in preoperative rehabilitation [14,30]. The situation has begun to change recently. In a study by Steinmetz et al., it was shown that a two-week preoperative exercise program (controlled aerobic training three times a week at a load intensity of 70% of VO2 peak) led to a significant increase in 6MWT distance, quality of life, and a decrease in Timed-Up-and-Go Test time [31]. This is consistent with the results of our study, in which in addition to increasing the strength of the stimulated muscles, it was also possible to obtain an increased 6MWT distance during the repeat test.

Now, a study is being carried out with longer preparation for surgery: pre-frailty and frailty patients will train for 6–10 weeks before planned coronary bypass surgery (two sessions per week—aerobics and resistance exercises) [32]. However, the patient’s condition does not always allow such a long course of pre-rehabilitation, and not all patients can safely undergo aerobic training due to the severity of their clinical condition. In such cases, the NMES course, similar to that used by us, can be applied. However, in pre-rehabilitation before cardiac surgery in the elderly, a significant problem is overcoming barriers in the participation of patients in such programs [31,33]. One way to solve this problem can be home-based training using NMES. This experience is available for patients with heart failure [34] and can also be extended to patients before cardiac surgery, as our study showed. Since the expectation of serious heart surgery (examination and preparation) can last at least two weeks, it is advisable to use NMES for outpatient rehabilitation. Even such a short course of respiratory muscle training reduced the number of respiratory complications after surgery [35].

### Study Limitations

The main limitation of this study is the lack of blinding of the study population and investigators. Unfortunately, the very nature of the NMES procedure is such that the study participant has a full understanding of whether he is being stimulated with muscle contraction or not. Additionally, the design of a single-centre study does not allow for the blinding of investigators. Therefore, the results of a prehabilitation program before cardiac surgery should be interpreted with caution. Although an identical approach was used in pre- and post-prehabilitation measurements of muscle status, we cannot exclude that the results of muscle strength assessment in the present study may overestimate the effect of NMES. To clarify this, a blinded, randomized controlled trial will be required to investigate the effectiveness of NMES in increasing muscle strength in patients before cardiac surgery. However, the results of this study indicate the advisability of NMES before cardiac surgery in this category of patients.

In addition, the duration of the NMES course may have been too short, but the relatively short preparation time of the patient in the hospital did not allow for an increase in the rehabilitation period. A longer course of NMES can likely be utilised before the operation on an outpatient basis, but since our research was carried out in a hospital, its results cannot be extended to an outpatient course of NEMS.

## 5. Conclusions

A short-term NMES course in patients awaiting cardiac surgery is feasible, safe, and effective in improving preoperative functional capacity (six-minute walk distance) and the maximal strength of stimulated muscles. It is questionable as to whether an increase in muscle strength and the functional state of patients after a course of NMES before surgery can improve the immediate results of cardiac surgery, and so the degree of recovery after NMES-supported cardiac surgeries requires further research.

## Figures and Tables

**Figure 1 ijerph-20-02678-f001:**
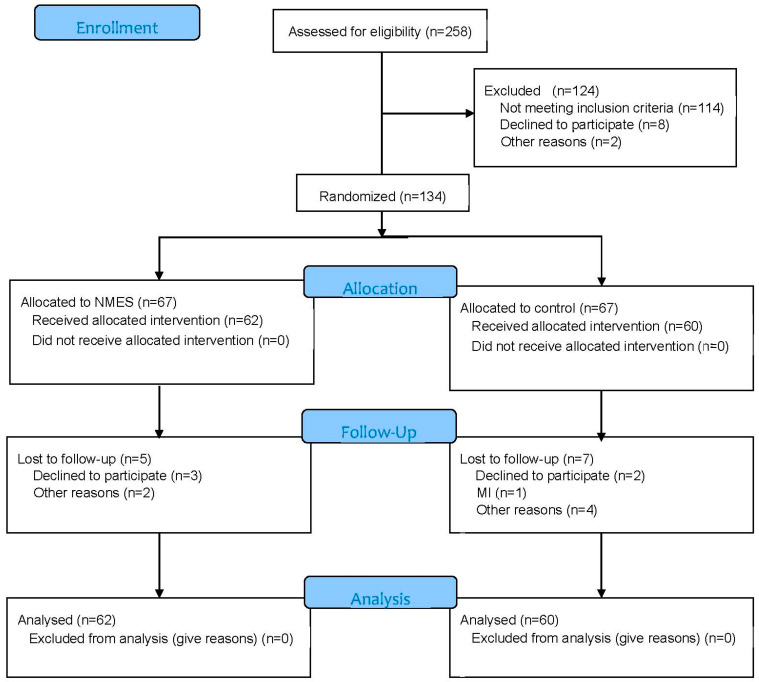
Flow diagram of study participants.

**Figure 2 ijerph-20-02678-f002:**
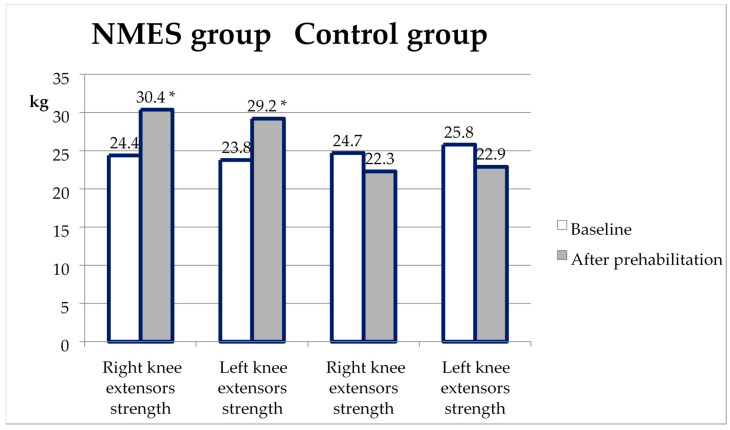
Changes in knee extensor strength from baseline to the second test in the NMES group and control. *—*p* < 0.05 compared with the control.

**Figure 3 ijerph-20-02678-f003:**
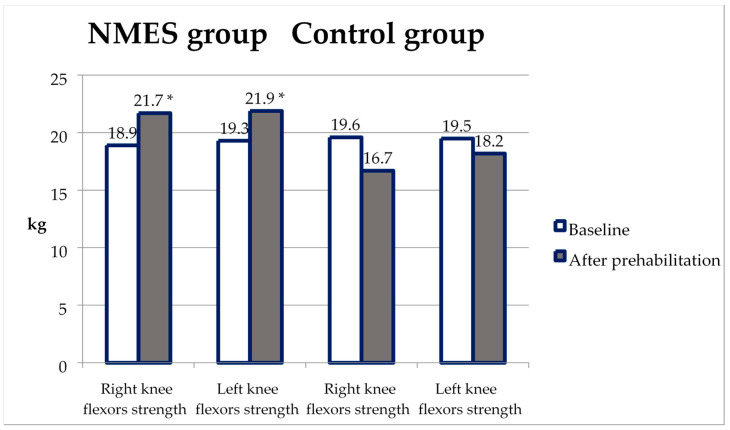
Changes in knee flexor strength from baseline to the second test in the NMES group and control. *—*p* < 0.05 compared with the control.

**Figure 4 ijerph-20-02678-f004:**
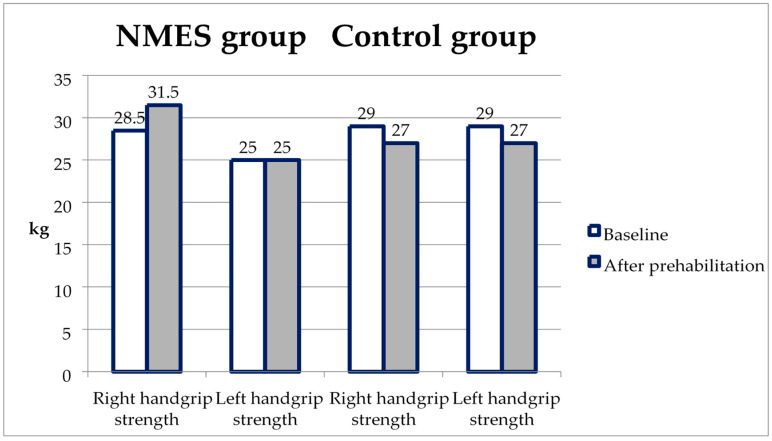
Changes in handgrip strength from baseline to the second test in the NMES group and control.

**Figure 5 ijerph-20-02678-f005:**
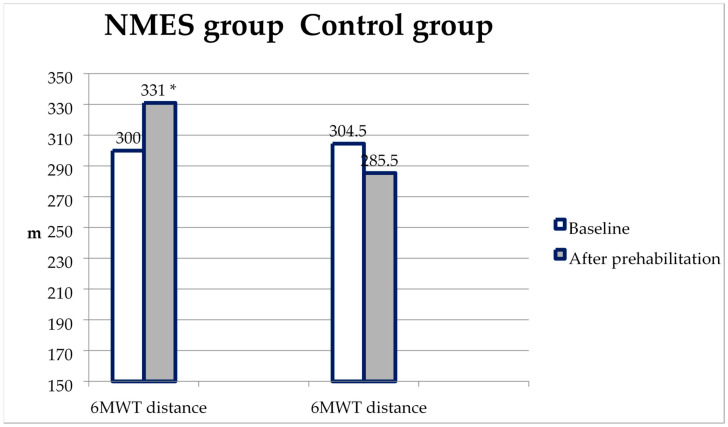
6 MWT distance changes from baseline to the second test in the NMES group and control. *—*p* < 0.05 compared with the control.

**Table 1 ijerph-20-02678-t001:** Baseline characteristics of patients.

	NMES Group (n = 62)	Control Group (n = 60)	*p* Value
Men (n, %)	44 (71.0)	39 (65.0)	0.339
Age (years)	62.0 [57.5; 66.6]	63.5 [59.0; 69.0]	0.131
Body mass index (kg/m^2^)	27.4 [25.4; 31.5]	28.7 [25.9; 33.3]	0.198
FC angina pectoris ≥ 3 (n, %)	16 (25.8)	20 (33.33)	0.337
Myocardial infarction history (n, %)	31 (50.0)	30 (50.0)	0.485
GFR (mL/min/1.73 m^2^)	84.5 [65.5; 94.5]	78.0 [64.0; 97.0]	0.778
Hypertension (n, %)	53 (85.5)	47 (78.3)	0.654
PCI history (n, %)	6 (9.7)	7 (11.7)	0.709
Stroke history (n, %)	5 (8.1)	8 (13.3)	0.313
Permanent atrial fibrillation (n, %)	10 (16.1)	8 (13.3)	0.748
Diabetes mellitus (n, %)	22 (35.5)	15 (25.0)	0.280
Peripheral arterial disease (n, %)	10 (16.1)	10 (16.7)	0.840
COPD (n, %)	10 (16.1)	5 (8.3)	0.214

**Table 2 ijerph-20-02678-t002:** Baseline characteristics of patients (laboratory and instrumental tests).

	NMES Group (n = 62)	Control Group (n = 60)	*p* Value
Left Atrial Diameter (cm)	4.5 [4.2; 5.0]	4.6 [4.2; 5.2]	0.387
Left Ventricle End-Systolic Volume (mL)	74.0 [47.0; 113.0]	64.0 [47.0; 97.0]	0.245
Left Ventricle End-Diastolic Volume (mL)	180.0 [135.0; 231.0]	173.0 [141.0;209.0]	0.636
Interventricular Septal Thickness (mm)	1.0 [1.0; 1.2]	1.0 [1.0; 1.3]	0.378
Posterior wall thickness (mm)	1.0 [1.0; 1.2]	1.0 [1.0; 1.2]	0.657
Aorta (cm)	3.5 [3.4; 3.8]	3.5 [3.3; 3.7]	0.221
Left ventricular ejection fraction (%)	60.0 [48.0; 66.0]	63.0 [48.0; 67.0]	0.142
Pulmonary artery systolic pressure(mmHg)	29.5 [23.0; 35.0]	32.0 [24.0; 45.0]	0.308
E/A ratio	0.69 [0.62; 1.06]	0.77 [0.66; 1.02]	0.306
Mitral regurgitation ≥ grade 3 (n, %)	8 (12.9)	6 (10.0)	0.686
Mitral stenosis (n, %)	3 (4.8)	8 (13.3)	0.084
Aortic valve regurgitation ≥ 3 grade (n, %)	4 (6.5)	1 (1.7)	0.202
Aortic stenosis (n, %)	8 (12.9)	9 (15.0)	0.656
Tricuspid regurgitation ≥ grade 3 (n, %)	4 (6.5)	7 (11.7)	0.274
Internal carotid artery stenosis ≥50% (n, %)	10 (16.1)	19 (31.7)	0.035
Chronic lower limb ischemia (n, %)	12 (19.4)	9 (15.0)	0.529
Glucose (mmol/L)	5.8 [5.3; 6.5]	5.8 [5.4; 6.3]	0.619
Creatinine (μmol/L)	81.0 [71.0; 93.0]	82.0 [70.0; 97.0]	0.934

**Table 3 ijerph-20-02678-t003:** Muscle strength of the lower and upper extremities and 6MWT during treatment time.

		NMES Group (n = 62)	Control Group (n = 60)
Right knee extensors strength (kg)	Baseline	24.4 [18.3; 31.4]	24.7 [20.1; 33.2]
	After prehabilitation	30.4 [23.8; 36.2] ^b^	22.3 [18.9; 30.4] ^b^
Left knee extensors strength (kg)	Baseline	23.8 [19.3; 31.3]	25.8 [19.2; 31.3]
	After prehabilitation	29.2 [23.6; 35.4] ^b^	22.9 [18.9; 27.8] ^b^
Right knee flexors strength (kg)	Baseline	18.9 [13.3; 24.0]	19.6 [13.1; 26.0]
	After prehabilitation	21.7 [16.6; 25.1] ^a^	16.7 [12.1; 23.3] ^a^
Left knee flexors strength (kg)	Baseline	19.3 [14.3; 24.5]	19.5 [13.0; 24.3]
	After prehabilitation	21.9 [17.3; 26.7] ^a^	18.2 [13.4; 22.2] ^a^
Right handgrip strength (kg)	Baseline	28.5 [20.5; 34.0]	29.0 [19.0; 34.0]
	After prehabilitation	31.5 [22.0; 34.0]	27.0 [19.0; 33.0]
Left handgrip strength (kg)	Baseline	25.0 [18.0; 31.0]	24.0 [15.0; 31.0]
	After prehabilitation	25.0 [18.0; 32.0]	22.0 [14.0; 28.0]
6MWT distance (m)	Baseline	300.0 [261,0;371,0]	304.5 [253.0; 380.0]
	After prehabilitation	331.0 [280,0;375,0] ^a^	285.5 [246.0; 342.0] ^a^

^a^—*p* < 0.01 between NEMS and control groups; ^b^—*p* < 0.001 between NEMS and control groups.

## Data Availability

Data regarding this manuscript are available in the Federal State Budgetary Scientific Institution “Research Institute for Complex Issues of Cardiovascular Disease”, Kemerovo, Russia.

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
