# Peer review of "Prehabilitation in Cardiovascular Surgery: The Effect of Neuromuscular Electrical Stimulation (Randomized Clinical Trial)"

_ijerph, 2023, doi:10.3390/ijerph20032678_

Round 1

Reviewer 1 Report

Dear authors,

I find this paper very interesting. It is a very relevant and pertinent topic.

I liked the discussion and the conclusions are deduced from the results.

In addition, the limitations exposed become strengths.

I have several doubts that I hope the authors will resolve.

Why were patients between 35 and 80 years of age taken?

I think the sample size is sufficient, but was a sample size study done - if so, I would like to know. If no sample size study was done I would like to know what the statistical power is.

 Minor revisión

The legend in the table caption of table 3 is not understood because it is repeated: a and b say the same thing.

Correct the titles and legends of the figures.

Author Response

Reply to Reviewer 1

Dear authors,

I find this paper very interesting. It is a very relevant and pertinent topic.

I liked the discussion and the conclusions are deduced from the results.

In addition, the limitations exposed become strengths.

I have several doubts that I hope the authors will resolve.

We would like to thank the referee for the favorable feedback on our article and useful comments that will improve the manuscript.

Why were patients between 35 and 80 years of age taken?

We included in the study patients aged 35 to 80 years who underwent preoperative preparation before cardiac surgery. Due to the existing cardiac pathology and severe coronary or heart failure, these patients have a decrease in their functional state. As a result, these patients develop secondary sarcopenia, a deterioration in muscle status, which we affected in the course of prehabilitation. Therefore, we did not limit the age at which patients were included in the study.

I think the sample size is sufficient, but was a sample size study done - if so, I would like to know. If no sample size study was done I would like to know what the statistical power is.

We did not calculate the required sample size because, to date, studies with NMES prior to cardiac surgery have not been performed. However, since we obtained statistically significant results, it seems that the number of included patients was sufficient. Thus, the statistical power ranged from 0.5 to 0.81 when comparing variables between the NMES and control groups in Table 3.

 Minor revisión

The legend in the table caption of table 3 is not understood because it is repeated: a and b say the same thing.

We have amended the text of the legend under table 3

 Correct the titles and legends of the figures.

We have corrected the names and legends of the Figures.

Reviewer 2 Report

Dear Authors

Thank you for the possibility to read and revise your manuscript entitled "Prehabilitation in Cardiovascular Surgery: The Effect of Neuromuscular Electrical Stimulation.

The study is interesting and provides novel insights into the potential applications of NMES in cardiovascular surgery. In general, the article is well written but there are some spelling and grammar mistakes that you should correct. 

Author Response

Dear Authors

Thank you for the possibility to read and revise your manuscript entitled "Prehabilitation in Cardiovascular Surgery: The Effect of Neuromuscular Electrical Stimulation.

The study is interesting and provides novel insights into the potential applications of NMES in cardiovascular surgery. In general, the article is well written but there are some spelling and grammar mistakes that you should correct. 

We would like to thank the referee for the favorable feedback on our article and useful comments that will improve the manuscript.

Reviewer 3 Report

This study aimed to assess the effect of prehabilitation with neuromuscular electrical stimulation (NMES) on muscle status and exercise capacity in patients before cardiac surgery.  The researchers used a clinical randomized study by comparing patients who underwent muscle stimulation daily before cardiac surgery with control patients who received the usual pre-surgery program (breathing exercises and education). The researchers concluded that a short-term NMES course before cardiac surgery is feasible, safe, and effective to improve preoperative functional capacity and strength of stimulated muscles. Here are my comments and suggestions:

·       The study design (RCT) is not mentioned in the title.  I suggest adding it

·       How do you explain the decreases in all outcomes of interest among the control group?

·       The group-by-time interaction should be tested {for example, by using mixed ANOVA, with one between-groups factor (treatment assignment) and one within-groups factor (time)}.  

·       In the control group, along with the preparation for the surgery, consisting of breathing exercises and an educational program, patients were recommended to adhere to their usual physical activity. Were patients in the NMES group also recommended to adhere to their usual physical activity?

·        In the abstract, please specify that the numbers 7-10 refer to sessions

·       In the introduction, I suggest placing greater focus on the association between prehabilitation with NMES and post-cardiovascular surgery outcomes

·       The study population consisted of individuals aged 35 to 85. It is possible that the intervention was more effective for older patients. I suggest assessing the impact of NMES at different age groups.

·       Please explain how sample size was determined

·       What is the difference between footnote a and footnote b in table 3?

Author Response

Reviewer 3

This study aimed to assess the effect of prehabilitation with neuromuscular electrical stimulation (NMES) on muscle status and exercise capacity in patients before cardiac surgery.  The researchers used a clinical randomized study by comparing patients who underwent muscle stimulation daily before cardiac surgery with control patients who received the usual pre-surgery program (breathing exercises and education). The researchers concluded that a short-term NMES course before cardiac surgery is feasible, safe, and effective to improve preoperative functional capacity and strength of stimulated muscles. Here are my comments and suggestions:

 We would like to thank the reviewer for the favorable feedback on our article and useful comments that will improve the manuscript.

  • The study design (RCT) is not mentioned in the title.  I suggest adding it

We added "Randomized Clinical Trial" to the title of the manuscript.

  • How do you explain the decreases in all outcomes of interest among the control group?

Indeed, during the preoperative preparation, there was a slight decrease in the functional state of skeletal muscles in control group (the decrease was not statistically significant for any indicator). This can be explained by the limitation of normal physical activity while in the hospital during preoperative preparation.

  • The group-by-time interaction should be tested {for example, by using mixed ANOVA, with one between-groups factor (treatment assignment) and one within-groups factor (time)}.  

Probably, indeed, such an analysis makes sense in order to simultaneously assess the influence of two factors and to test group interaction over time. However, we did not originally plan to conduct such an analysis when registering the study protocol on the Clinicaltrias.gov website (NCT04545268). When planning the next studies, we will definitely take into account your remark and will plan this type of analysis as well.

  • In the control group, along with the preparation for the surgery, consisting of breathing exercises and an educational program, patients were recommended to adhere to their usual physical activity. Were patients in the NMES group also recommended to adhere to their usual physical activity?

Yes, for patients in the NMES group, we also recommended regular physical activity. Another thing is that the stay in the hospital inevitably caused some limitation of physical activity (due to the presence of symptoms of the underlying disease, the patients were not inclined to physical activity).

  • In the abstract, please specify that the numbers 7-10 refer to sessions

We corrected the text of the abstract

  • In the introduction, I suggest placing greater focus on the association between prehabilitation with NMES and post-cardiovascular surgery outcomes

Thank you for your suggestion, however, we cannot discuss this issue in the introduction, since so far there have been no studies on prehabilitation with the help of NMES and evaluation of the outcomes of heart surgery. We are conducting such a study in our group, the results will be presented in subsequent publications. We discuss the clinical perspectives of NMES prehabilitation in the Discussion section, as well as in the Conclusions section.

  • The study population consisted of individuals aged 35 to 85. It is possible that the intervention was more effective for older patients. I suggest assessing the impact of NMES at different age groups.

We included in the study patients aged 35 to 80 years who underwent preoperative preparation before cardiac surgery. Due to the existing cardiac pathology and severe coronary or heart failure, these patients have a decrease in their functional state. As a result, these patients develop secondary sarcopenia, a deterioration in muscle status, which we affected in the course of prehabilitation. Therefore, we did not limit the age at which patients were included in the study. In the future, we plan to study the effect of NMES purposefully in patients of the older age group.

  • Please explain how sample size was determined

We did not calculate the required sample size because, to date, studies with NMES prior to cardiac surgery have not been performed. However, since we obtained statistically significant results, it seems that the number of included patients was sufficient. Thus, the statistical power ranged from 0.5 to 0.81 when comparing variables between the NMES and control groups in Table 3.

  • What is the difference between footnote a and footnote b in table 3?

We have amended the text of the legend under table 3

Round 2

Reviewer 1 Report

The authors have responded to all my comments. I have no further comments. Thank you

Reviewer 3 Report

Thanks to the authors for addressing my comments. I recommend accepting the paper. Good luck!